# Research Progress and Development Demand of Nanocellulose Reinforced Polymer Composites

**DOI:** 10.3390/polym12092113

**Published:** 2020-09-17

**Authors:** Renjie Shen, Shiwen Xue, Yanru Xu, Qi Liu, Zhang Feng, Hao Ren, Huamin Zhai, Fangong Kong

**Affiliations:** 1Jiangsu Provincial Key Lab of Pulp and Paper Science and Technology, Nanjing Forestry University, Nanjing 210037, China; 3191500481@njfu.edu.cn (R.S.); shiwen@njfu.edu.cn (S.X.); yanru@njfu.edu.cn (Y.X.); qliu@njfu.edu.cn (Q.L.); 141501204fz@njfu.edu.cn (Z.F.); hzhai@njfu.edu.cn (H.Z.); 2State Key Laboratory of Biobased Material and Green Papermaking, Qilu University of Technology, Shandong Academy of Sciences, Jinan 250353, China; kfg@qlu.edu.cn

**Keywords:** nanocellulose, biomass, polymers, composite, functionalization

## Abstract

Nanocellulose is a type of nanomaterial with high strength, high specific surface area and high surface energy. Additionally, it is nontoxic, harmless, biocompatible and environmentally friendly and can be extracted from biomass resources. The surface groups of cellulose show high surface energy and binding activity on the nanoscale and can be modified by using various methods. Because nanocellulose has a high elastic modulus, rigidity and a low thermal expansion coefficient, it is an excellent material for polymer reinforcement. This paper summarizes the reinforcement mechanisms of nanocellulose polymer composites with a focus on the role of theoretical models in elucidating these mechanisms. Furthermore, the influence of various factors on the properties of nanocellulose reinforced polymer composites are discussed in combination with analyses and comparisons of specific research results in related fields. Finally, research focus and development directions for the design of high-performance nanocellulose reinforced polymer composites are proposed.

## 1. Introduction

Biomass, as a raw material, has emerged in response to the need for sustainable development. With the depletion of fossil resources and to meet economic development and population growth, the industrial production of new materials from renewable biomass has become a key objective [1]. Since Nickerson and Habrle first prepared nanocellulose from wood by acidolysis in 1947 [2], nanomaterials obtained from plant resources with various application prospects have attracted great attention [3,4]. Nanocellulose is the natural fiber which can be extracted from cellulose, the main structural component of plant cell walls. Nanocellulose shows promise as a new type of reinforcing filler in the field of composite materials [5,6]. However, the types of raw materials as well as the separation and extraction methods affect its physical and chemical properties. Presently, there are many studies and applications, technical problems such as preparation of nanocellulose, the dispersion and interfacial compatibility between nanocellulose and the polymer mobile phase remain, resulting in high cost and underdeveloped industrialization of nanocellulose composites [7]. This paper attempts to summarize existing research on nanocellulose reinforced polymer composites, combining theoretical models and reinforcing mechanisms, to determine a breakthrough point and directions for future research.

This section reviews the basic characteristics of nanocellulose from three aspects: definition, preparation and properties.

### 1.1. Definition

Cellulose is a material consisting of linear polymeric chains of β-D-glucopyranose units linked by glycosidic bonds at their C1 and C4 positions [8,9]. In this paper, we adopt the definition reported in “ISO/TS 20477: 2017 Nanotechnologies—Standard terms and their definition for cellulose nanomaterial,” that is, the term nanocellulose (NC) is synonymous with cellulose nanomaterials, which is further divided into cellulose nanoobjects and cellulose nanostructured materials. Cellulose nanoobjects are cellulose materials with one, two or three external dimensions on the nanoscale. This kind of material include cellulose nanocrystals (CNCs) and cellulose nanofibrils (CNFs). Cellulose nanostructured materials are cellulose materials with internal or surface nanostructures, including cellulose microcrystals (CMCs) and cellulose microfibrils (CMFs). Table 1 summarizes the sizes of these different kinds of nanocellulose [10].

### 1.2. Preparation

Cellulose is one of the most abundant natural polymers on Earth. It is the main reinforcing phase of the biologic structure of plants and can also be synthesized by using algae and certain bacteria. Therefore, there are many kinds of raw materials available for the preparation of nanocellulose, including wood [11,12], bacterial cellulose [13], processed plant fiber products such as pulp [14,15], waste materials (mainly includes straw and bagasse [16,17,18]) and microcrystalline cellulose (MCC) [19,20,21], which are commonly used in laboratory research. Various methods have been used to prepare nanocellulose from these materials [22]. There are two main extraction processes in the preparation of nanocelluloses: chemical (such as acid hydrolysis [23,24,25] and enzymatic hydrolysis [19,20]) and mechanical treatments (including milling, high-pressure homogenization and ultrasonication [26,27]). Methods that combine chemical and mechanical treatments (such as high-pressure homogenization coupled with strong acid hydrolysis pretreatments [28] or ball mill pretreatment combined with maleic acid hydrolysis [29]) were found to generate high quality products [30]. The preparation process has a considerable influence on the surface structure of nanocellulose. A specific process often results in a specific surface property. For example, hydrochloric acid imparts the surface with a large number of hydroxyl groups [31], while sulfuric acid hydrolysis results in a sulfonated surface and the degree of sulfonation can be adjusted by controlling the acid concentration, temperature and time [32]. Acetylated surfaces can be obtained by a Fischer–Speier esterification with acetic acid [33]. The sulfonated surface obtained by the sulfate method is negatively charged, which is conducive to the dispersion of nanocellulose to form a stable suspension, and thus, it is the most common method. Ionic liquids and other new methods for preparing nanocellulose have also been developed in recent years [11,34]. The diversity of raw materials and preparation methods results in the diverse structure, morphologies and properties of nanocellulose.

Mechanical methods, without oxidation and acidolysis, usually result in a hydroxylated surface similar to that of natural cellulose [26,27]. The increasingly popular chemo-mechanical method entails 2,2,6,6-tetramethylpiperidine-1-oxyl (TEMPO) oxidation combined with mechanical treatment [35]. Pretreatment with TEMPO oxidation can reduce cellulose degradation and allow for milder mechanical treatment conditions. The obtained nanocellulose has a carboxylic surface, which can also form a stable suspension when negatively charged by ionization. Although mechanical separation of plant fibers into smaller components usually requires a higher energy input, chemical or enzymatic preprocessing of fibers has been developed to overcome this problem [35,36]. Due to the initiation of industrial research activities are underway [37], industrial production lines for the preparation of nanocellulose are currently in operation [38,39].

### 1.3. Properties

Nanocellulose has high crystallinity. The cellulose molecules are arranged in a regular line and the intra- and intermolecular hydrogen bonds make cellulose a relatively stable polymer and impart the cellulose fibers with high axial stiffness and strength to cellulose fibers [40]. The tensile strength of nanocellulose is much higher than that of steel wire and general carbon fiber, reaching 7.5–7.7 GPa, while the density (1.6 g/cm^3^) is only one-fifth of that of steel [8]. Moreover, nanocellulose has a high specific surface area with a large number of exposed hydroxyl groups, which is beneficial for the grafting or crosslinking of other substances [41] to obtain different surface properties for surface functionalization [42]. Chemical modification of the surfaces of nanocellulose increases its dispersibility of nanocelluloses within organic solvents or matrices [43,44]. In addition, because of its high specific surface area, the adsorption capacity of nanocellulose is high, which can be further enhanced by chemical modification. Therefore, it has great potential as adsorption materials in purification and filtration membranes and other fields [45,46,47]. Each type of nanocellulose (CNC, CNF, CMC and CMF) brings new properties. Moreover, nanocellulose, especially CNC, has the advantages of high strength, high rigidity, low density, low thermal expansion coefficient, biocompatibility and renewable sources [48]. Because of these advantages, increasing attention was paid to the application of nanocellulose in composite materials. Its potential applications include engineering plastics [49,50], biodegradable plastics [51,52], biomedical materials [53,54,55], water purification material [46,47,56] and electronic component materials [57,58,59].

## 2. Surface Modification

Because of the compatibility problem between hydrophilic nanocellulose and most hydrophobic polymers, surface modification of nanocellulose is often necessary to obtain better reinforcement. The chemical structure of the nanocellulose surface determines its dispersion effect, composite manufacturing process and the property of the resultant composite products. It also provides more options for the performance adjustment of composite materials. Currently, numerous surface treatment technologies for nanocellulose have been developed. These modification methods can be divided into two main categories: electrostatic adsorption and chemical modification.

### 2.1. Electrostatic Adsorption

Due to the mostly hydrophilic surface, nanocellulose does not disperse easily in organic media and polymer solutions, so surfactants are needed to stabilize them. Nanocellulose obtained by sulfuric acid hydrolysis or pretreated by TEMPO oxidation has a charged surface for the adsorption of surfactants; commonly used surfactants are stearic acid and cetyl-tetramethyl-bromide-amine [60]. The compatibility of nanocelluloses with an organic medium and matrix is enhanced after treatment. It is also conducive to the dispersion of nanoparticles [61]. In addition to surfactants, many studies have used inorganic metal ions to perform specific functions [62,63]. For example, nanocellulose coated with tannic acid coordinated silver ions to form a sterilized and self-adhesive surface [64].

Referencing wet-end chemistry used in papermaking, another common method is to use polyelectrolytes such as linear cationic polyacrylamide (CPAM) and branched polyethylenimine (bPEI) [57]. As mentioned earlier, nanocellulose can form a stable suspension by ionization, while polyelectrolytes affect adsorption and flocculation, by the actions of charge neutrality action and polyelectrolyte bridging action [65], resulting in the formation of a supramolecular structure with specific functions [66]. This is also the mechanism of the solution coagulation method.

### 2.2. Chemical Modification

Chemical modification can be used to create a surface with a specific function by attaching other molecules using covalent or ionic bonds. There is a large amount of hydroxyl groups on the surface of nanocellulose, and thus, substances that react with alcohols such as isocyanates, epoxides, acid halides and anhydrides, are commonly used reactants. These substances form a series of specific surface groups such as amine, quaternary ammonium salt, alkyl, hydroxyl alkyl, alkoxy, ester and acid [67]. These surface chemical modifications can increase the dispersibility of nanocellulose in organic solvents and their compatibility with the polymer matrix [68]. Chemical modifications also make it possible to directly attach nanoparticles directly to the polymer matrix by covalent bonding without the use of coupling agents or surfactants [69,70,71].

Grafting is another method used to attach polymers to the surface of nanocellulose. For example, polycaprolactone (PCL) biocomposites, which have become a research focus in recent years, can be prepared by grafting PCL (acting as the compatibilizer) on the surface of nanocellulose as a compatibilizer using hydroxyl radicals as the initiator and catalyst, in order to prepare biodegradable nanocellulose reinforced polycaprolactone composites [72].

## 3. Reinforcing Mechanism of Nanocellulose

The modification of nanocellulose should ultimately serve the purpose of improving the properties of the composites. To explain how the abovementioned modification methods influence the reinforcing effect of nanocellulose in polymer materials, the reinforcing mechanism of nanocellulose should be further studied.

A nanoparticle-reinforced polymer is essentially a dynamic problem. Theoretically, this process can be simulated at the molecular level by modeling with using simulation software. Moon et al. [10] systematically studied the reinforcing mechanism of nanocellulose. First, molecular mechanics (MM) and molecular dynamics (MD) were used for modeling. The existing mathematical simulation model of cellulose was comprehensively analyzed to study the interaction between nanocellulose and other materials. The research focused on the interaction function, energy distribution and orientation of the nanocellulose surface with other materials. Based on these studies, they summarized a mathematical model for describing the mechanical behavior of nanocellulose reinforced polymer composites. The model can predict the upper and lower bounds of the properties of the composites under certain conditions and mathematically revealed the reinforcing mechanism of nanocellulose reinforced polymer composites. It therefore provides important guidance for the research of new high-performance nanocellulose reinforced polymer composites. Next, the strengthening mechanism is summarized and analyzed from several aspects.

### 3.1. Properties of the Matrix and Nanocellulose

The properties of nanocellulose (reinforcing phase) and the polymer (matrix) will considerably affect the properties of the composite products. Properties associated with the enhancement phase mainly include the axial elastic modulus (E_A_), axial shear modulus (G_A_), axial Poisson’s ratio (ν_A_), transverse volume modulus (K_T_) and transverse shear modulus (G_T_). Matrix correlation properties are mainly the volume modulus (K_m_) and shear modulus (G_m_) [44]. These indices determine the upper and lower limits of the mechanical properties of the composites, including the volume modulus (K_c_) and shear modulus (G_c_). As it is assumed that nanocellulose is well distributed with random orientation, a mathematical expression can be obtained as follows [73]:(1)$${\left(\frac{V_{f}}{K_{T}}+\frac{V_{m}}{K_{m}}\right)}^{-1}\leq K_{c}\leq \frac{V_{f}}{9}\left(E_{A}+4K_{T}{\left(1+\nu_{A}\right)}^{2}\right)+V_{m}K_{m},$$
(2)$$\begin{array}{l} {\left(\frac{V_{f}}{15}\left(\frac{1}{K_{T}}+6\left(\frac{1}{G_{A}}+\frac{1}{G_{T}}\right)+\frac{3\left(1+4\nu_{A}\right)}{E_{A}}\right)+\frac{V_{m}}{K_{m}}\right)}^{-1}\\ \leq G_{c}\leq \frac{V_{f}}{15}\left(E_{A}+6\left(G_{A}+G_{T}\right)+K_{T}{\left(1-2\nu_{A}\right)}^{2}\right)+V_{m}G_{m} \end{array}$$
where V_f_ and V_m_ represent the volume fraction of the reinforcement phase and the matrix, respectively. The experimental data showed that in most cases, regardless of the state of the two phases in the phase diagram, the potential properties of the composite production will be constrained within half an order of magnitude of the determined upper and lower bounds. On this basis, Halpin and Kardos posited a mean field theory to comprehensively consider the upper and lower bounds [74]. The model takes the weighted average of the upper and lower bounds according to the distribution and arrangement of the nanometer reinforcing phase to obtain a more accurate prediction.

### 3.2. Dispersion and Orientation

As mentioned earlier in 3.1, the range of elementary upper and lower bounds is relatively wide and the mean field theory established to obtain more accurate predicted values depends on the dispersion and orientation of the reinforcing phase in the matrix. The experimental results showed that for the same enhancement system, the different distribution and orientation of the reinforcing phase will lead to a large fluctuation in the performance of the composite [75].

The distribution of nanoparticles involves two important factors, the size and the size distribution of the nanoparticles. If the size is too large, the nanometer effect is weakened, which includes a decrease in the number of phase interfaces, a weakening of the bond strength between the reinforcing phase and the matrix and a deterioration of the stress conduction effect [44]. If the size is too small, another problem is presented, which is the increased difficulty in the preparation and dispersion of the nanoparticles. Although many of the theoretical models of nanocellulose reinforced polymer composites require the size of the nanoparticles to be as small as possible, for nanocellulose, due to the high surface activity and hydrophilicity of nanocellulose, self-aggregation is inevitable without a certain amount of processing, it easily self-aggregates, which makes dispersion more difficult to disperse, and the actual effective particle size cannot reach the theoretical value. If the particle size distribution is too wide, the short plate effect will be prominent, and the enhancement effect will be reduced. On the other hand, increasing the compatibility between the reinforcing phase and the matrix greatly affects the dispersion. According to Halpin’s mean field theory model [74], an increased dispersion of nanocellulose is indicative of a higher “proportion” of the upper bound in the model, implying better performance of the composite materials can be obtained.

In addition to the distribution mode, the arrangement mode of nanocellulose also has a significant influence on the properties of the composites. Nanocellulose is a type of nanomaterial with a high aspect ratio in most cases, composed of linear molecules, and its axial and radial mechanical properties are considerably distinct. Its orientation and arrangement inevitably affect the properties of the composite materials [75,76]. In addition to the three-dimensional random distribution mentioned above, the mathematical models of the 2D and 1D distributions were also studied and analyzed. Among them, the “concentric cylinder” 1D distribution model proposed by Hashin is the most rigorous [77]. Unlike 3D materials, 2D composites show high strength only in the distribution plane and in the 1D model, in the axial direction of the nanocellulose arrangement [78]. By comparing the three models, it can be seen that increasing the orientation degree of nanocellulose also increases the elastic modulus of the composite material in a specific direction increases by 3–4 times [10].

### 3.3. Interface

The interface is another factor that affects the matrix reinforcing effect of nanocellulose. Its effects include stress conduction in the matrix and fracture limit in the stress–strain process [79].

For a single nanocellulose, the properties that affect the interface include its shape (aspect ratio) and interface stiffness coefficient (*Ds*). The latter is an important indicator of interfacial properties in that *Ds* = 0 indicates a completely discontinuous or debonding interface and *Ds* = ∞ indicates a perfect interface without discontinuity [42]. The detailed mathematical analysis of the effects of the two on the properties of the composites can be simplified by replacing the axial and shear moduli of nanocellulose with the effective modulus [79]:(3)$${E_{A}}^{*}=E_{A}\left(1-\frac{tanh\beta_{\kappa }}{\beta_{\kappa }}\right),$$
(4)$${G_{A}}^{*}=\frac{G_{A}}{1+\frac{2G_{A}}{r_{f}D_{s}}},$$
where *β* is the stress conduction rate. According to the shear lag theory, in addition to being affected by the properties of the reinforcing phase and the matrix itself, *β* is positively correlated with the interface stiffness coefficient. Therefore, the model predicts that the development of effective nanocellulose composites will depend on an interface with a high *Ds* or more physically on an interface that can conduct stress from the matrix to the nanocellulose reinforcing phase as quickly as possible [80].

The mathematical model is relatively reliable and accurate for the analysis of the material properties of the material in the elastic stage. However, in the toughness stage, the viscous flow strain of the matrix is more complicated and is greatly affected by stress loading conditions. Therefore, research on the fracture properties of nanocellulose composites is limited, and mainly focuses on the microscopic image and fracture energy of the fractured surface.

The fracture of a composite material has many potential mechanisms such as matrix fracture, fiber fracture, interfacial debonding, particle cavitation, crack path change and shear zone [81]. Some studies have shown that the interface has a key effect on the fracture properties of the composites [44]. On one hand, nanocellulose is regarded as a rigid particle. According to the toughening mechanism, the stress concentration is due to the deformation of the uniformly distributed stress field, creating a matrix yield that absorbs more deformation work. On the other hand, a matrix segmented by the interface can effectively inhibit further development of the microcrack, improving the rupture limit of the material [81]. Since the fracture of the composite can be traced back to plastic deformation at the tip of the microcrack, the additional energy expended by the interface debonding and the yield of holes in the matrix are the main reasons for the toughening. Therefore, the goal of toughening nanocomposites is to develop an energy dissipation system that can absorb more energy by controlling the interface properties of nanoparticles.

### 3.4. Percolation Concepts

In the process of analyzing existing research results, Moon R et al. observed experimental data beyond the prediction results of Halpin’s mean field theory. This part of the transcendent phenomenon exists objectively; in order to explain this phenomenon, they referred to the concept of percolation in a heterogeneous system and presented the concepts of percolation in nanocellulose composites [82].

In a uniform and stable matrix, the distribution of the stress field is also uniform. However, the relatively high rigidity of nanocellulose induces deformation of the stress field and the force field extends along the two-phase interface. When the distribution of nanocellulose exceeds a certain degree, the percolation of nanocellulose into each other forms a microscopic network and produces a synergistic effect. In addition to the volume fraction of the reinforcing phase, the formation of the network is also related to the aspect ratio, the dispersion and the arrangement of nanocellulose. Further research showed that an improvement of the aspect ratio and better dispersion were conducive to a reduction in the percolation threshold [83], facilitating the synergistic effect and a performance leap to be achieved at a lower nanocellulose volume fraction, that is, a wider selection range of nanocellulose content.

The significance of percolation networks in composite materials is that when the volume fraction of the reinforcing phase exceeds the percolation threshold, some properties of the material will improve sharply, and it may even achieve properties that previously did not exist may be achieved. Therefore, the density and dispersion of nanoparticles in the matrix must be carefully controlled to obtain a good percolation network.

In summary, existing theoretical studies have provided relatively accurate mathematical models for simulation; however, in practical applications, many parameters in the model, especially the performance of nanocellulose, cannot be accurately measured, and it is difficult to measure changes during the process of preparation and modification, which introduces errors to the theoretical value. There are still some obstacles to the practical application of the theoretical models. To solve these problems, it is necessary to improve the preparation and modification process of nanocellulose and develop more accurate methods of characterization and measurement.

## 4. Preparation of Nanocellulose Reinforced Polymer Composites

As mentioned above, the upper and lower limits of the composite material properties are determined by the reinforcing phase and the properties of the matrix themselves. However, the specific level of the prepared composite properties is closely related to the internal structure of the material. The contradiction in the processing of nanocellulose reinforced polymer matrix composites is that both good nanocellulose dispersion and a nanocellulose network structure with controllable contact degree should be formed. To achieve the optimal structures and obtain the composite materials with the best comprehensive performance, the following issues will need to be addressed during the surface modification of nanocellulose and the preparation process of composite materials: dispersion, orientation and arrangement of nanocellulose; combination with the matrix; the number and properties of interfaces; and other indices. In the following sections, existing research ideas are analyzed from the most commonly used processes for preparing nanocellulose reinforced polymer matrix composites—such as solution casting, solution coagulation, melt blending, electrostatic spinning and sol–gel solvent exchange.

### 4.1. Solution Casting

Nanocellulose or modified nanocellulose is predispersed in a specific medium. Thereafter, a polymer solution is added to disperse it and then compressed into a compact form after evaporation or freeze drying to separate the solvent, which is often used to prepare nanocellulose composite films [84]. However, the solvent often restricts the application scope of the solution casting method. Due to the hydrophilicity of nanocellulose, this method is often used for composite materials with water-soluble or water-dispersive polymers as the matrix. Common polymer matrices include polyvinyl alcohol (PVA) [85,86] and waterborne polyurethane (WPU) [87,88]. Other studies have shown that nanocellulose can be dispersed in N,N-dimethylformamide (DMF) and the stability of the dispersion is the same as that in aqueous solutions [89]. Therefore, DMF is often used as the dispersant in addition to water.

The solution casting method is often used to prepare nanocomposite films. In addition to the thermal and mechanical properties, the barrier properties of the film are important as they directly affect the application prospects of composite films. In terms of technology, research on the preparation of nanocomposite films by the solution casting method should focus on two aspects. First, compatibility between nanocellulose and the matrix should be improved. Because both nanocellulose and the matrix are “forced” to disperse by the solvent; if the compatibility between nanocellulose and the matrix is too poor, it is likely that nanocellulose will self-assemble rather than connect with the matrix during solvent evaporation. Second, it is necessary to design a suitable dispersion and solvent treatment system. Due to the needs of industrialization, improper solvent selection may lead to high recovery costs and environmental pollution.

#### 4.1.1. Water as Solvent

Water is one of the most common dispersants of nanocellulose. As water is a solvent that is most feasible for solvent recovery and pollution control, it is an ideal dispersant for the preparation of nanocellulose composite films by solution casting [85,86]. Water can also be used as a dispersant of polyvinyl alcohol (PVA), which has excellent mechanical properties, a facile film formation ability and a strong binding affinity with cellulose through hydrogen bonding [90]. The mechanical properties and thermal stability of PVA can be significantly improved by adding nanocellulose. However, evaporation of the solvent during solution casting can easily lead to the agglomeration of nanocellulose, which seriously reduces the enhancement effect. Therefore, it is necessary to focus on the development of a system that can disperse nanocellulose in PVA and other matrices during solution casting.

Miri et al. [85] studied a new functional hybrid nanomaterial consisting of cellulose nanocrystals (CNC) and graphene oxide nanosheets (GONs) and successfully prepared PVA nanocomposites with different mass fractions by solution casting with water as the dispersing medium. Studies found that GON and CNC generated synergistic effects through interactions (as shown in Figure 1a,b), which effectively prevented the agglomeration of nanoparticles in the polymer matrix, improved the dispersion uniformity and greatly improved the mechanical properties, thermal stability and moisture absorption of the nanocomposite. Aloui et al. [86] used CNC and halloysites (HNTs) in the same way to obtain mixed nanofillers with similar synergistic effects and prepared their composite materials with PVA. In addition to the improvement in mechanical properties, the material also exhibited good water vapor barrier and thermal stability.

#### 4.1.2. DMF as Solvent

N,N-dimethylformamide (DMF) is a very useful solvent that can be mixed with water and most organic solvents, except halogenated hydrocarbons. Moreover, it is chemically stable, is a good solvent for a variety of organic compounds and inorganic compounds and has been found to be an excellent solvent for solution casting and film forming. Compared with water, DMF is the better solvent and can dissolve both hydrophilic and hydrophobic polymers. However, when hydrophobic polymers are used as the matrix, due to the poor compatibility between nanocellulose and hydrophobic polymers, the research focus should be on improving the interface [42].

Cao et al. [87] prepared a series of novel waterborne polyurethane (WPU)-cellulose nanocrystal (CNC) composites. By using polycaprolactone (PCL) as the compatibilizer, the partially presynthesized WPU chain was grafted on the surface of CNCs, and the corresponding nanocomposite was obtained by casting and evaporation of DMF as the solvent. Experimental results show that in WPU–CNC composites, CNC surface grafting of the WPU chain and PCL soft segments jointly formed the crystalline domains; a eutectic was formed between the substrate and the packed continuous phase, which significantly enhanced the interface adhesion of CNC, thereby improving the thermal stability and mechanical strength of the nanocomposites. Similarly, Seoane et al. [91] prepared poly (3-hydroxybutyrate)-based CNC biocomposites by the solution casting method using DMF as the solvent. Similar accelerated crystallization was observed. The results show that the composite material has good resistance to water vapors and ultraviolet rays, in addition to excellent mechanical properties.

### 4.2. Solution Coagulation

Solution casting is not a suitable method for water-insoluble biodegradable polyesters. In the study of some nanocellulose composites involving biodegradable polyesters, the solution coagulation method was used, that is, the solute was precipitated by adding a poor solvent into the well-dispersed predispersions as a coagulant, so as to separate the solvent and obtain the composite products. Unlike solution casting, solution coagulation can effectively prevent nanoparticles from accumulating because it avoids evaporation over a long period of time. Li et al. [92] prepared poly(butyl succinate) (PBS)/cellulose nanocrystal (CNC) composites by solution coagulation with DMF as the matrix solvent, water as the strengthening solvent and excess water as the coagulant. The experimental results showed that, compared with pure PBS, the crystallization temperature and overall crystallization rate of the nanocomposite increased significantly due to heterogeneous nucleation. When the CNC content was 1.0 wt%, the tensile strength and modulus increased by 20% and 10.5%, respectively.

### 4.3. Melt-Compounding

Melt-compounding is a method in which nanofillers are directly added to the molten polymer and dispersed by mechanical action and then directly extruded to produce the products (such as granulation and wire rod). As the most plausible industrial preparation process of nanocellulose composites, melt-compounding has attracted considerable research. The process is generally used in the manufacture of engineering materials, and related research is focused on the dispersion effect of nanocellulose, thermal and mechanical properties, crystallization and the thermal, mechanical and rheological properties of the composite materials [49].

Although melt blending is the most convenient process with the lowest cost, the mechanical dispersion of nanocellulose in molten polymers leads to the degradation of nanocellulose by the melting temperature and high shear strength, which may change the crystal structure and thus affect the interface compatibility [93].

Biodegradable, high-strength polylactic acid (PLA) derived from biomass sources may be the most promising material to achieve the industrial production of a wide range of engineering materials. However, due to its hardness, brittleness and low thermal stability, PLA still has some problems in the processing and application of PLA exist. An effective way to improve the thermal, mechanical and processing properties of PLA is by adding nanocellulose as the reinforcing phase.

#### 4.3.1. Unmodified Nanocellulose

The industrial production, the production process usually needs to be as short as possible to reduce cost and improve production efficiency. However, the modification of nanocellulose usually requires fine control and relatively high cost. Therefore, it is proposed that the industrial production of nanocellulose products is industrially produced without complex modifications and directly used for producing composite materials.

Jonoobi et al. [94] studied the preparation of cellulose nanofiber (CNF)-reinforced PLA by melt-compounding. To achieve uniform dispersion and full mixing of nanofillers, they first used a mixed solution of acetone and chloroform as the solvent. CNF and PLA premixed powders were prepared by solution casting. The premixed powders were then placed in a twin-screw extruder as a filler in the molten PLA; dried, mixed, diluted and granulated; and finally, polylactic acid-nanocellulose composites were obtained as industrial products. It was found that 5% CNF can create composites with the best thermal and mechanical properties. Kamal et al. [95] adopted another method. By spray freeze drying, they obtained porous bulk cellulose (CNCSFD) with a micro–nano network from which PLA-based CNCSFD composites with good dispersion were prepared by melt-compounding. Further research showed that the network structure of CNCSFD improved the low-frequency rheological properties (loss modulus, storage modulus and composite viscosity) of PLA through interactions between CNC particles. Moreover, as a nucleating agent, CNC significantly improved the crystallization rate of the composite during isothermal or nonisothermal crystallization.

#### 4.3.2. Modified Nanocellulose

Although composites of unmodified nanocellulose and PLA composite can be obtained by adjusting the production process, as an effective means to adjust the compatibility and interface performance of the interface between nanocellulose and the matrix, modification of nanocellulose has incomparable advantages over other methods, such as better dispersion, stronger combination and functionalization [61].

Raquez et al. [96] greatly improved the compatibility between nanocellulose and the matrix by silylation of nanocellulose in a citrate buffer solution. After drying, they were melt-compounded with PLA. FTIR, TEM, WAXS and XPS analyses were used to demonstrate the effectiveness of this method in the surface modification of nanocellulose. Spinella et al. [49] modified the surface of nanocellulose crystals by the Fischer esterification method to obtain different surfaces depending on the type of acetate esterification (acetate or lactate). Thus, the surface characteristics of modified nanocellulose can easily be controlled and adjusted by chemical modification. The thermal stability temperature of the modified nanocellulose was increased by 40 °C, and thus, modified nanocellulose PLA composites were prepared by the melt-compounding method did not result in degradation of nanocellulose. Compared with pure PLA, under and above vitrification temperature, the storage modulus of modified nanocellulose-reinforced PLA increased by 31% and 450% under and above the vitrification temperature, respectively. Further research showed that this modification method can effectively improve the dispersion of nanocellulose in the PLA matrix.

Melt-compounding, as the most suitable method for the industrial mass production of composite materials, has solved the problem of dispersion and compatibility of nanocellulose in PLA to some extent; however, the material manufacturing cost is still relatively high and further research and development of new technologies are needed to achieve industrial mass production.

### 4.4. Electrostatic Spinning

By applying an electric field to the polymer solution or melt, the droplets can be stretched into nanoscale fibers under the action of static electricity. Nonwoven nanofibers are collected. However, with the recent development in electrostatic spinning technologies, an improved receiver can obtain highly directionally aligned nanofibers. Its significance lies not only in the fine control of the nanoscale structure, but it can also in the achievement of directional arrangement of nanocellulose in the composites. This effect, as described in Section 3.2, can greatly improve the axial strength of the composites. Further studies have found that the nonwoven nanofiber pads produced by this technology are closer to the components of the extracellular matrix components than those using traditional technologies and have potential applications as biologic tissue scaffolds [97]. Moreover, the nanofibers obtained by electrostatic spinning have many advantages, such as high specific surface area, adjustable porosity and the control of nanofiber components to achieve desired properties and functions. Moreover, the fiber mat formed by composite fibers on the collecting device can form films directly, which makes electrostatic spinning a strong candidate in developing nanocomposite films with specific functions.

Although it is also necessary to predisperse nanoparticles in a molten polymer or solution, it is easier to achieve uniform dispersion of nanoparticles in nanofibers by the electrostatic spinning method than solution casting and fusion mixing because of the extremely fast rate of solvent separation or melt cooling. Moreover, because of the mild treatment conditions, degradation of the crystal structure of nanocellulose was effectively avoided and more effective interfaces were retained. Therefore, the focus of research is on improving the interface compatibility between nanocellulose and the matrix at the interface.

#### 4.4.1. Biologic Tissue Scaffolds

To study the application of nanocellulose reinforced polymer composites as biologic tissue scaffolds, Zhou et al. [53] prepared biodegradable and cellular-compatible composite fiber bionanomaterials with good mechanical properties by electrostatic spinning using maleic anhydride (MAH)-grafted PLA as the matrix and CNC as the reinforcing phase. MAH grafting of PLA improved the hydrophilicity of PLA and enhanced the interaction between CNC and the matrix. In addition, the axial orientation effect of nanocellulose generated by electrostatic spinning further improved the stability of the material under high-strength loads. In addition, with PLA as the matrix, Zhang et al. [54] successfully synthesized PEG-grafted CNCs, which were added to PLA as the reinforcing filler to prepare PEG–PLA nanocomposite scaffold materials by electrostatic spinning. Experiments showed that the interface binding between CNC and PLA was improved by grafting PEG onto CNC. Moreover, the tensile strength of the composite fiber was effectively improved by adding 5% PEG-grafted CNC. The aforementioned studies demonstrated that nanocellulose reinforced polymer composites have good biocompatibility and can be used as biologic tissue scaffolds.

#### 4.4.2. Nanocomposite Films

Arrieta et al. [51] prepared flexible films by electrospinning biodegradable nanocomposites in flexible films. They first blended polylactic acid (PLA), polyhydroxybutyrate (PHB) and acetyltributylcitrate (ATBC) into flexible fibers, and then dispersed 1 wt% and 5 wt% cellulose nanocrystal (CNC) into the system. Finally, flexible biologic nanocomposite films were obtained by electrostatic spinning. The addition of CNC improved the thermal resistance and mechanical resistance of the fiber and imparted an appropriate surface water resistance. In addition, all the raw materials of the biocomposite can be completely decomposed under composting conditions. Thus, due to these combined properties, the films can be used in agriculture as flexible compostable materials.

The main advantage of electrostatic spinning is the achievement of nanofibers with highly ordered and controllable structures. Moreover, it has the potential to realize various functionalization of the materials. This potential is further enhanced by the addition of nanocellulose, which can be modified by multiple methods. Presently, the various possibilities of new composite materials made of different functionalized nanocellulose with different forms of fiber forms realized by electrostatic spinning require further exploration.

### 4.5. Sol–Gel Method

The sol–gel method (S–G method for short) converts a colloidal solution (sol) into a gel-like network (gel) with improved mechanical properties, which are further enhanced by heat treatment. Although this process is time-consuming and uses harmful solvents, it is still a research focus for the preparation of functional nanocomposites in the laboratory because it can guarantee uniform dispersion on the molecular level under mild treatment conditions.

Using polycaprolactone (PCL) and polyethylene glycol (PEG) as soft segments and cellulose nanocrystals (CNC) as crosslinkers, Liu et al. [69] developed a shape-memory polymer comprising nanocomposite networks and exhibiting thermal and water responsiveness by the sol–gel method (Figure 2). The nanocomposite exhibited excellent thermo-induced and water-induced shape-memory effects in water at 37 °C (close to body temperature) and it is biocompatible. The introduction of CNC significantly improved the mechanical properties of the low molecular-weight PEG and PCL polymers. This study demonstrated the application value of the sol–gel method in the preparation of nanocellulose reinforced polymer composites with complex microstructures, although further studies are required to optimize the efficiency of the materials preparation.

## 5. Summary and Prospect

Nanocellulose reinforced polymer composites have been studied in many fields. Although the research emphasis, methods and results of these studies are not the same, they all have the same ultimate goal—to design functional materials with high performance and high added-value by using the following advantages of nanocellulose: lightweight, high strength, low cost and renewable sources. While nanocellulose reinforced polymer composites have some advantages in performance, barriers to their marketization and industrialization exist and some unknown areas need further exploration.

### 5.1. Advantages of Nanocellulose Reinforced Polymer Composites

The modification of polymers has been studied since Staudinger founded Polymer Science and as long as the demand for high-performance materials remains, the subject will continue to flourish. Thus, because the market potential of nanocellulose reinforced polymer composites is huge, an accumulation of scientific research achievements exists; many of these achievements that have been translated into practical applications. Compared with traditional modification methods such as blending modification, nanofiller-reinforced polymers have the ability to improve the strength of the material while not significantly reducing or even enhancing the toughness of the material, which provides more possibilities for the application of the material and thus a stronger competitiveness.

As mentioned above, the following characteristics of nanocellulose make it an ideal nanofiller: extremely low particle size, high specific surface area, high modulus, high strength and ease of modification. If the existing preparation processes with high pollution and energy consumption can be further improved, it can also be a completely natural biomass, which is widely sourced, biodegradable, nontoxic, biocompatible and environmentally friendly. All of these factors make nanocellulose an important choice in the fields of biodegradable plastics, bioscaffolds, catalysts and drug carriers.

### 5.2. Current Challenges

Nanocellulose reinforced polymer composites have development potential and several applications. However, there are still two problems that must be overcome to promote its industrial application and utilization—functionalization and mass production.

#### 5.2.1. Functionalization

There is an inherent compatibility problem between hydrophilic nanocellulose and most hydrophobic polymer matrices, which greatly affects the dispersion of nanocellulose and its combination with the matrix. Most of the abovementioned research progress listed in the previous article involved the modification of nanocellulose, but in fact, improving the interfacial compatibility is only one of the objectives. Modified nanocellulose can lead to a more diverse composite design and improve performance. This is both an opportunity and a challenge. Owing to the unique surface properties of nanocellulose, there are many ways to modify it. What needs to be considered is not only the purpose of the modification, but also the influence of the modification on the particle morphology, surface microstructure, crystalline structure and physical and chemical properties of nanocellulose and the influence of these changes on the performance of the final composite, which needs further research to improve. In addition, numerous studies have shown that the synergistic effect of nanocellulose and other nanofillers results in more complex effects. Functionalization strategies that are different from the two-phase system of nanocellulose and uniform matrix, requires the establishment of a multiphase model, which is difficult because the reinforcement mechanisms have not yet been fully elucidated, which needs further study.

On the other hand, researchers also face the challenge of developing modified materials with higher value-added materials. Improvements in the properties of most nanocellulose reinforced polymer composites are based on the original properties of the polymers. However, in recent years, there have been a large number of improved nanocellulose composite materials with properties (such as conductive [57,58,59], barrier [87,98], shape memory [69], compostable [51]) that were not originally available to the constituents individually. Thus, the research goal has gradually changed from improving the properties of existing materials to creating new functional materials to meet increasingly specialized needs, thereby introducing another challenge for the research of nanocellulose reinforced polymer composites.

#### 5.2.2. Mass Production

To realize mass production, the most important problem to address is the production cost. Due to its high production cost and low production capacity, the price of nanocellulose is several times higher than other common nanofillers such as nano-montmorillonite and nanosilica. In addition, modification or other pretreatment are often needed before compounding with the polymer, which further increases the production cost of the nanocellulose-reinforced polymer, hampering the marketization of current nanocellulose reinforced polymer composites. This is the main problem hindering the mass production process. Optimizing the process to reduce production costs and increase production capacity to achieve batch production is the biggest challenge facing mass production applications.

As described in Section 1.2, owing to the wide range of sources and various production methods of nanocellulose, the performance of nanocellulose products supplied as raw materials is uneven, making it is difficult to standardize the product index. In all the specific studies cited in this paper, the nanocellulose materials that were used, even if they are of the same type, have some order of difference in their crystallinity, particle morphology, particle size distribution and other properties. The raw materials, methods and process parameters of nanocellulose preparation will have different degrees of impact on the nanocellulose products and thus, affect the properties of the composite products. Thus, further development of the preparation of nanocellulose on a large scale is required to control this fluctuation in properties within an acceptable range. At the same time, although there have been many significant studies [99], further research is required to explore efficient, sensitive and unified means to characterize the properties of nanocellulose or modified nanocellulose, especially the percentage of crystallinity, the location of the amorphous region, the ratio of I_α_/I_β_ crystal forms and the identification of crystal defects. Almost all contemporary reports on the mechanical properties of a specific nanocellulose fall within a wide range, which will hinder the establishment of accurate models and performance predictions. Thus, theoretical research, which relies heavily on mathematics, will fail to keep up with the latest research progress. Research and development on new batch homogenization production methods and complete measurement standards will play an important role in promoting the construction of accurate measurement systems and standards for the mechanical properties of single nanocellulose.

### 5.3. Future Development

In summary, the two directions for the development of nanocellulose reinforced polymer composites are functionalization and mass production, which essentially caters to the demand for better, stronger, cheaper and more variety in materials. In fact, many of the current challenges represent the development potential of this field. Regardless of theory, technology and application, any small breakthrough may propel the whole industry toward a major milestone. Nanocellulose reinforced polymer composites are playing an increasingly important role in the following applications: engineering plastics, functional films, bioscaffolds, catalysts, drug carriers, adsorption materials and flexible electronic components, with more potential applications under investigation. Thus, with further research, nanocellulose reinforced polymer composites are expected to become the most valuable composite material in the coming era.

## Figures and Tables

**Figure 1 polymers-12-02113-f001:**
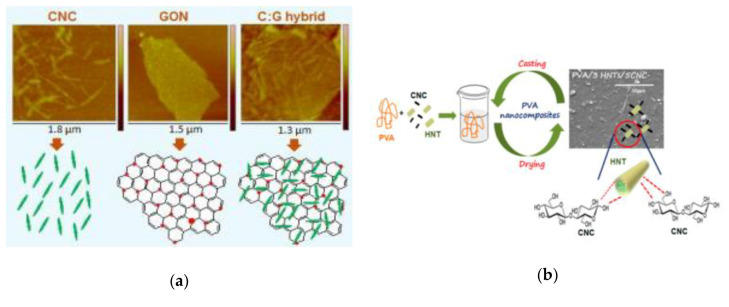
(**a**) Electron microscopy image and schematic diagram of CNC, GON and C: G (C:G = 1:2) mixed particles (red dot: oxygen-containing group in GON). Reprinted with permission from [85]. Copyright 2016 Elsevier; (**b**) structural diagram of HNTs-CNC. Reprinted with permission from [86] Copyright 2016 American Chemical Society.

**Figure 2 polymers-12-02113-f002:**
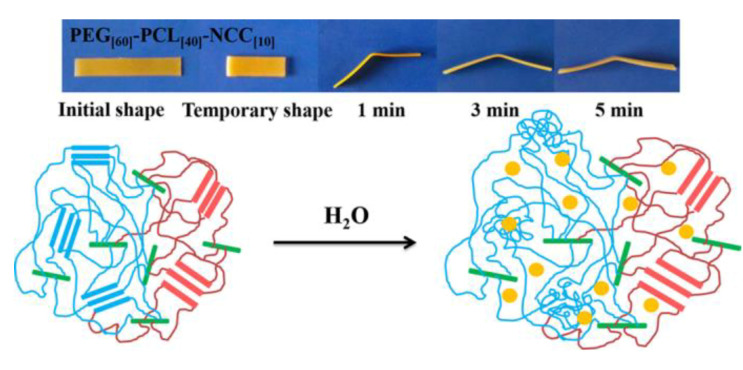
PEG_[60]_–PCL_[60]_–NCC_[10]_ shape-memory effect and its mechanism. Reprinted with permission from [69]. Copyright 2015 American Chemical Society.

**Table 1 polymers-12-02113-t001:** Summary of several kinds of nanocellulose.

Particle Type	Length (μm)	Width (nm)	Height (nm)	Aspect Ratio	Form
CNC	0.05–0.5	3–5	3–5	5–50	Pure crystalline structure
CNF	0.5–2	4–20	4–20	>50	Contain both crystalline regions and amorphous regions
CMC	10–50	10–50 (μm)	10–50 (μm)	<2	Manufactured by partially depolymerizing high purity cellulose
CMF	0.5–10	10–100	10–100	>50	Contain multiple elementary fibrils with both crystalline regions and amorphous regions

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
