# Peer review of "Research Progress and Development Demand of Nanocellulose Reinforced Polymer Composites"

_polymers, 2020, doi:10.3390/polym12092113_

Round 1

Reviewer 1 Report

The manuscript polymers-920928 review the reinforcement mechanisms of nanocellulose polymer composites with focus on the role of theoretical models in elucidating these mechanisms.

I recommend the publication of this manuscript in Polymers journal, after minor revisions.

  • The manuscript needs extensive revision by a native English speaker!

Abstract

  • What the authors meant by “high surface activity”? Could it be energy?? Please revise the sentence!
  • “nanocellulose is rigid, has a high elastic modulus, rigidity”? Please revise the sentence!
  • Please change “The surface groups of cellulose … can be surface-modified using various methods” with “The surface groups of cellulose … can be modified by using various methods”!
  • Please change “the influence of various factors that influence the properties of nanocellulose reinforced polymer composites” with “the influence of various factors on the properties of nanocellulose reinforced polymer composites”!

Introduction

  • Please change: “With the depletion of fossil resources, the industrial production of new materials from renewable biomass to meet economic development and population growth has become a prominent objective” with “With the depletion of fossil resources and to meet economic development and population growth, the industrial production of new materials from renewable biomass has become a key objective”!
  • Please change: “Nanocellulose is usually derived from the cell walls of natural plant fibers.” with “Nanocellulose is the natural fiber which can be extracted from cellulose, the main structural component of plant cell walls.”!
  • The sentence “the type of raw materials, the separation and extraction methods affect its physical and chemical properties” appear twice in the Introduction section!!!
  • Please change: “polymer reinforcing mechanisms” with “reinforcing mechanisms”!

  1. Introduction to nanocellulose” not “Introduction of nanocellulose”!
  • “Therefore, there are many kinds of raw materials are available for the preparation”?? Please change with “Therefore, there are many kinds of raw materials available for the preparation”!
  • Please revise the sentence: “Therefore, there are many kinds of raw materials are available for the preparation of nanocellulose, such as wood[10] or bacterial cellulose[11], processed plant fiber products such as pulp[12], waste materials such as straw or bagasse[13], and microcrystalline cellulose (MCC)[14,15], which are commonly used in laboratory research.”! There is a disturbing repetition of “such as”!!
  • There are two main extraction processes in the preparation of nanocelluloses: chemical (acid hydrolysis) and mechanical treatments. Please make this visible in the sentence “Various methods have been used to prepare nanocellulose from these materials, such as acid hydrolysis[16,17] and enzymatic hydrolysis [14,15], and mechanical methods, such as milling, high-pressure homogenization, and ultrasonication [18,19].”!!
  • The sentence “such as milling, high-pressure homogenization, and ultrasonication [18,19]” is repeated at pag 3, thus, should be deleted!
  • Please clarify the sentence “Methods that combine chemical and mechanical treatments”! Please mention between brackets the chemical methods used in preparation of nanocelluloses!
  • What did the authors want to say about: “high-speed shearing”?? The mechanical disintegration is obtained by using high-pressure homogenizer, grinders, microfluidizers, cryo-crushing and high-intensity ultrasonication. Please revise this paragraph!

2.1. Electrostatic adsorption

  • What did the authors want to say with “formation of a supermolecular structure”?? Maybe a supramolecular structure???

Author Response

Response to reviewer:

Thank you for your guidance and suggestions. We have carefully referred to your suggestion and revised our manuscript again. Here please allow me to respond to the comments point by point.

  1. What the authors meant by “high surface activity”? Could it be energy?? Please revise the sentence!

  We have revised “high surface activity” to “high surface energy” in Abstract.

  1. “nanocellulose is rigid, has a high elastic modulus, rigidity”? Please revise the sentence!

  I'm sorry for our misexpression, we have revised “nanocellulose is rigid, has a high elastic modulus, rigidity” to “Moreover, nanocellulose has a high elastic modulus, rigidity” in Abstract.

  1. Please change “The surface groups of cellulose … can be surface-modified using various methods” with “The surface groups of cellulose … can be modified by using various methods”!

  We have revised “The surface groups of cellulose … can be surface-modified using various methods” to “The surface groups of cellulose … can be modified by using various methods” in Abstract.

  1. Please change “the influence of various factors that influence the properties of nanocellulose reinforced polymer composites” with “the influence of various factors on the properties of nanocellulose reinforced polymer composites”!

  We have revised “the influence of various factors that influence the properties of nanocellulose reinforced polymer composites” to “the influence of various factors on the properties of nanocellulose reinforced polymer composites” in Abstract.

  1. Please change “With the depletion of fossil resources, the industrial production of new materials from renewable biomass to meet economic development and population growth has become a prominent objective” with “With the depletion of fossil resources and to meet economic development and population growth, the industrial production of new materials from renewable biomass has become a key objective”!

  We have revised “With the depletion of fossil resources, the industrial production of new materials from renewable biomass to meet economic development and population growth has become a prominent objective” to “With the depletion of fossil resources and to meet economic development and population growth, the industrial production of new materials from renewable biomass has become a key objective” in Introduction.

  1. Please change: “Nanocellulose is usually derived from the cell walls of natural plant fibers.” with “Nanocellulose is the natural fiber which can beextracted from cellulose, the main structural component of plant cell walls.”!

  We have revised “Nanocellulose is usually derived from the cell walls of natural plant fibers.” to “Nanocellulose is the natural fiber which can be extracted from cellulose, the main structural component of plant cell walls.” in Introduction.

  1. The sentence “the type of raw materials, the separation and extraction methods affect its physical and chemical properties” appear twice in the Introduction section!!!

  We deleted the first of the repeated sentences that“the type of raw materials, the separation and extraction methods affect its physical and chemical properties”in Introduction, and only the latter sentence that “However, the types of raw materials as well as the separation and extraction methods affect its physical and chemical properties. ” is retained.  

  1. Please change: “polymer reinforcing mechanisms” with “reinforcing mechanisms”!

  We have revised “polymer reinforcing mechanisms” to “reinforcing mechanisms” in Introduction.

  1. “Introduction to nanocellulose” not “Introduction of nanocellulose”!

  We have revised “Introduction of nanocellulose” to “Introduction to nanocellulose” in the title of section 1.

  1. “Therefore, there are many kinds of raw materials are available for the preparation”?? Please change with “Therefore, there are many kinds of raw materials available for the preparation”!

  We have revised “Therefore, there are many kinds of raw materials are available for the preparation” to “Therefore, there are many kinds of raw materials available for the preparation” in section 1.2.

  1. Please revise the sentence: “Therefore, there are many kinds of raw materials are available for the preparation of nanocellulose, such as wood[10] or bacterial cellulose[11], processed plant fiber products such as pulp[12], waste materials such as straw or bagasse[13], and microcrystalline cellulose (MCC) [14,15], which are commonly used in laboratory research.”! There is a disturbing repetition of “such as”!!

  I’m sorry for our poor English expression that there are too many "such as" in this paragraph. We have revised this sentence to “Therefore, there are many kinds of raw materials are available for the preparation of nanocellulose, including wood or, bacterial cellulose, processed plant fiber products such as pulp, waste materials (mainly includes straw or and bagasse), and microcrystalline cellulose (MCC), which are commonly used in laboratory research.” in section 1.2.

  1. There are two main extraction processes in the preparation of nanocelluloses: chemical (acid hydrolysis) and mechanical treatments. Please make this visible in the sentence “Various methods have been used to prepare nanocellulose from these materials, such as acid hydrolysis[16,17] and enzymatic hydrolysis [14,15], and mechanical methods, such as milling, high-pressure homogenization, and ultrasonication [18,19].”!!

  We have rivised this sentence to “There are two main extraction processes in the preparation of nanocelluloses: chemical (such as acid hydrolysis and enzymatic hydrolysis), and mechanical treatments, (including milling, high-pressure homogenization, and ultrasonication).” in section 1.2.

  1. The sentence “such as milling, high-pressure homogenization, and ultrasonication [18,19]” is repeated at pag 3, thus, should be deleted!
  2. What did the authors want to say about: “high-speed shearing”?? The mechanical disintegration is obtained by using high-pressure homogenizer, grinders,microfluidizers, cryo-crushing and high-intensity ultrasonication. Please revise this paragraph!

  There are sentence repetition and unclear expression, so we deleted the sentence “which include high-intensity ultrasound, high-pressure homogenization, and high-speed shearing, do not oxidize or degrade the products” and revised the sentence to “Mechanical methods, without oxidation and acidolysis, usually result in a hydroxylated surface similar to that of natural cellulose .”  on page 3, section 1.2.

  1. Please clarify the sentence “Methods that combine chemical and mechanical treatments”! Please mention between brackets the chemical methods used in preparation of nanocelluloses!

  Refer to the comments, we have revised this sentence to “Methods that combine chemical and mechanical treatments (such as high-pressure homogenization coupled with strong acid hydrolysis pretreatments[28] or ball mill pretreatment combined with maleic acid hydrolysis[29]) ” in section 1.2.

  1. What did the authors want to say with “formation of a supermolecular structure”?? Maybe a supramolecular structure???

  Here is our word mistake. We have revised  “formation of a supermolecular structure” to  “formation of a supramolecular structure” in section 2.1.

We do have many mistakes in expression. We have tried our best to improve our language expression and thank you again for your help.

Best wishes

Reviewer 2 Report

see attached file

Author Response

Response to reviewer:

Thank you for your guidance and suggestions. We have carefully referred to your suggestion and revised our manuscript again. 

  1. About “some imprecise opinions at the final part of the text” mentioned in the review report, we changed “At the same time, further research is required to…” to “At the same time, although there have been many significant studies, further research is required to…” in the original textin 5.2.2, and the recommended references (Foster et al 2018 review, Soc. Rev., 2018, 47,2609. DOI: 10.1039/c6cs00895j) are cited here.

  1. About “ lack of exhaustive and incomplete collection of references” mentioned in the review report, we referred to many document again and a total of 100 references were cited after this revision.In addition, we rearranged the list of references to avoid mistakes.

Thank you again for your help.

Best wishes

Round 2

Reviewer 2 Report

Please consider to change in ref. 46 the authors name was written as lastname, correct it.

Please check and re-check the references puntuation and spacing.